# IntegronFinder 2.0: Identification and Analysis of Integrons across Bacteria, with a Focus on Antibiotic Resistance in Klebsiella

**DOI:** 10.3390/microorganisms10040700

**Published:** 2022-03-24

**Authors:** Bertrand Néron, Eloi Littner, Matthieu Haudiquet, Amandine Perrin, Jean Cury, Eduardo P. C. Rocha

**Affiliations:** 1Bioinformatics and Biostatistics Hub, Institut Pasteur, Université de Paris Cité, 75015 Paris, France; bertrand.neron@pasteur.fr (B.N.); amandine.perrin@pasteur.fr (A.P.); 2Microbial Evolutionary Genomics, Institut Pasteur, Université de Paris Cité, CNRS UMR3525, 75015 Paris, France; eloi.littner@pasteur.fr (E.L.); matthieu.haudiquet@pasteur.fr (M.H.); 3DGA CBRN Defence, 91710 Vert-le-Petit, France; 4Collège Doctoral, Sorbonne Université, 75005 Paris, France; 5Ecole Doctorale FIRE–Programme Bettencourt, CRI, 75004 Paris, France; 6Laboratoire Interdisciplinaire des Sciences du Numérique, Université Paris-Saclay, CNRS UMR 9015, INRIA, 91400 Orsay, France

**Keywords:** integron, antibiotic resistance, bioinformatics, genomics

## Abstract

Integrons are flexible gene-exchanging platforms that contain multiple cassettes encoding accessory genes whose order is shuffled by a specific integrase. Integrons embedded within mobile genetic elements often contain multiple antibiotic resistance genes that they spread among nosocomial pathogens and contribute to the current antibiotic resistance crisis. However, most integrons are presumably sedentary and encode a much broader diversity of functions. IntegronFinder is a widely used software to identify novel integrons in bacterial genomes, but has aged and lacks some useful functionalities to handle very large datasets of draft genomes or metagenomes. Here, we present IntegronFinder version 2. We have updated the code, improved its efficiency and usability, adapted the output to incomplete genome data, and added a few novel functions. We describe these changes and illustrate the relevance of the program by analyzing the distribution of integrons across more than 20,000 fully sequenced genomes. We also take full advantage of its novel capabilities to analyze close to 4000 *Klebsiella pneumoniae* genomes for the presence of integrons and antibiotic resistance genes within them. Our data show that *K. pneumoniae* has a large diversity of integrons and the largest mobile integron in our database of plasmids. The pangenome of these integrons contains a total of 165 different gene families with most of the largest families being related with resistance to numerous types of antibiotics. IntegronFinder is a free and open-source software available on multiple public platforms.

## 1. Introduction

Bacteria evolve novel traits by many processes, including horizontal gene transfer (HGT) driven by phages, conjugative elements, or natural transformation. The impact of human activities has challenged bacteria in numerous ways, including antibiotic therapy and stress caused by pollution and this has spurred the adaptation of bacteria by HGT [1,2]. Once novel genes are acquired by the abovementioned processes, their integration in the novel host genome and subsequent expression is facilitated by the action of mobile genetic elements (MGEs) that rearrange genetic information within genomes [3,4]. Among elements reshaping bacterial genomes, integrons have a particularly important role in gene shuffling and allow the concentration of certain genes in compact genetic regions, which may result in the co-transfer of related functions (reviewed in [5]). The action of integrons also provides a genetic mechanism to quickly vary the expression of these genes [6]. As a result, integrons have become notorious drivers of the transfer and expression of antibiotic resistance genes [5].

Integrons are an assembly of several genetic elements that include an integrase (IntI) and a succession of gene cassettes flanked by recombination attachment sites (*att*) (Figure 1A). The integrase evolved from the typical tyrosine recombinases to mediate ssDNA recombination by the means of a folded substrate [7]. This unusual activity for a tyrosine recombinase is associated with an extra domain in the protein close to the C-terminus of the protein [7] that allows IntI to be easily distinguished from other integrases [8]. IntI can interact with recombination sites (attachment sites) and lead to the excision or integration of gene cassettes [7]. Integrons have typically one initial attachment site (*attI*) close to the integrase where integration of gene cassettes takes place. The cassettes are flanked by another type of site (*attC*) whose recognition by IntI leads to cassette excision. As a result, the integrase promotes the excision of internal cassettes in the integron and its integration at the *attI* site [9,10,11]. The *attC* sites are very variable in sequence but conserved in secondary structure [9]. Covariance models using this information can be used to find them efficiently [10]. Since near the *attI* site there is a promoter and most cassettes are devoid of promoters, the cassettes closest to the *attI* site are more likely to be expressed, even if internal promoters may also exist [11]. Hence, integrons include several cassettes and the action of the integrase promotes their shuffling, thereby varying those expressed at a given moment. 

One usually distinguishes sedentary from mobile integrons [6]. Sedentary integrons are encoded in the chromosome and can be very large, up to hundreds of cassettes in certain strains of *Vibrio* spp. [12]. In contrast, mobile integrons have few cassettes and are often contained in larger MGEs, such as plasmids. Mobile integrons have been grouped in five classes defined by their similarity in terms of the sequence of IntI, which seem to have emerged recently [13]. Class I integrons are particularly abundant and have the capacity of recombining more diverse types of cassettes (i.e., a broader variety of *attC* sites) than sedentary integrons [14]. This flexibility and their association with MGEs have probably contributed to the presence of many antibiotic resistance genes in integrons and their subsequent spread across nosocomial bacteria. Yet, not all clades of bacteria carry integrons. While these elements are very abundant in Gamma and Beta Proteobacteria [10,15,16], they are only very occasionally identified in strains of Alpha Proteobacteria, Actinobacteria, or Firmicutes [10,17], even among those that are also human-associated. The reasons for this remain unclear. While the frequency of antibiotic resistance genes in integrons has driven a lot of interest in these elements [18], it is likely that in natural environments, they are linked with other traits. The peculiar ability of integrons to shuffle genetic information is also being leveraged to develop synthetic biology approaches [19].

Six years ago, we published an extensive analysis of the distribution of integrons in bacteria [10]. In that work, we searched for integrons, but also for integron-integrases lacking cassettes (In0) and clusters of *attC* sites lacking integron-integrases (CALIN) (Figure 1B–D). For this, we developed a software called IntegronFinder that has become popular to identify integrons in bacterial genomes. The program searches for the integrase using hidden Markov models protein profiles with the software HMMer [20], and for the *attC* sites using covariance models with the software Infernal [21]. These and other genetic components, e.g., promoters, are then clustered if they are co-localized in the DNA sequence. IntegronFinder is thus able to identify integrons with novel cassettes and is complementary to the use of databases such as Integrall [22] that store known integrons and allow to search for homology of the cassettes by sequence similarity. The program is particularly useful to study the wealth of integrons found in environmental bacteria and which often encode unknown functions [23]. Another software to identify *attC* sites, HattCI, was made available at approximately the same time [24]. More recently I-VIP was published to identify class 1 integrons using sequence similarity to a database of integrases and our own covariance model [16]. However, no software seems to be currently maintained and developed. This is a problem because IntegronFinder version 1 has some limitations. Notably, it was designed to analyze complete genomes. The analysis of draft genomes and metagenomes was possible but required scripting skills and managing an output that was not designed for such data. This is a limitation since recent studies have uncovered novel integrons in metagenomes and in metagenome-assembled genomes [25,26]. IntegronFinder version 1 has also aged poorly in that it was written in Python 2 (now deprecated) and lacks modern tools to facilitate its maintenance. Finally, some of its annotation databases are no longer updated. We have therefore refactored the program, changed the outputs, added flexibility, and updated reference databases to come up with a novel version that is much better adapted to the analysis of incomplete genome data and to stand the passage of time (and pervasive lack of funding for software maintenance). Key changes are indicated in Figure 1E.

## 2. Materials and Methods

### 2.1. Refactoring of IntegronFinder

The first version of IntegronFinder was coded in Python v2.7, which is now deprecated. The program was ported to Python 3.7 (but currently works with versions up to Python 3.10) and the code was refactored to improve efficiency and correct minor bugs. Promoters and *attI1* sites are no longer detected by default to increase speed, especially as *attI* sites are poorly described (and hence not detected by IntegronFinder) outside class I integrons. We have added unit (or non-regression) tests, which make the software more robust and more attractive for future contributors.

IntegronFinder can now be easily run in parallel with a Nextflow script [27] that we provide ready to use. We have also diversified the installation methods, so it can be easily deployed on a variety of machines. Notably, we built a Singularity container which will allow a smooth installation on clusters. We have also updated and improved the documentation, especially on the developer part so that anyone can contribute to the code, to add novel features or fix bugs. 

### 2.2. Novel Functionalities

IntegronFinder v2 has some novel functions in relation to the previous version. A key novel functionality is the systematic use of the gathering thresholds (--cut_ga option in hmmer) that allows to identify hits to the protein profiles with better accuracy than arbitrary e-values. For this, we introduced a gathering threshold in the protein profile specific to the IntI delivered in IntegronFinder that was calculated using the hits of our previous analysis [10].

The reference database for annotating antibiotic resistance genes in the first version of IntegronFinder was RESFAMS (http://www.dantaslab.org/resfams, accessed on 11 November 2018) [28]. However, this database has not been updated recently and while it fits the needs of researchers in metagenomics aiming at identifying distant homologs of antibiotic resistance genes, it is less appropriate to identify antibiotic resistance genes identified in clinical samples. Hence, IntegronFinder now includes the AMRFinderPlus HMM profiles by default [29]. However, any other HMM database, including RESFAMS, can be input in the program. For example, one can input the entire PFAM database [30] for broader functional annotation of the cassettes (at the cost of significant increase in running time).

### 2.3. Input

IntegronFinder now accepts multifasta files as input. This is a major change since it allows the analysis of genome drafts or metagenome data without the need for scripting an analysis (i.e., calling IntegronFinder recurrently and managing the output). Draft genomes should be used with the linear replicon type (corresponding to the fact that they are contigs). By default, IntegronFinder uses the linear replicon type if there are multiple contigs in the input file. If there are multiple replicons in the input file with different topologies, this can be specified by providing a topology file. IntegronFinder also accepts an option --gembase that corresponds to a particular gene identifier that allows to put multiple replicons in the same file and inform the program that they should be regarded as different replicons. This format can be created by PanACoTA, a pipeline to automatically build the basic bricks of comparative genomics automatically, including download, annotation, and formatting of genome data [31]. The sequence files are initially annotated using Prodigal [32] to provide accurate and uniform annotations of all the sequences. 

As described above the input can also include files for annotation of the integron, e.g., for antibiotic resistance genes, using any kind of database compatible with hmmer (http://hmmer.org/documentation.html, accessed on 5 February 2022). Interestingly, a recent study has produced diverse *attC* covariance models for different clades [33]. These can be used in IntegronFinder by invoking the option --attc-model to use another covariance model instead of the one used by default.

### 2.4. Output

The output of IntegronFinder has been changed to integrate the use of draft genomes. Only three files are now created by default (see the “output” section of the documentation for details about the other possible output files). These three files will be sufficient for most researchers. They include the main output file that contains the position of the different elements (*attC* site, integrase, CDS, etc.). A summary file contains the information of the number and type of elements (complete, In0, and CALIN, see Figure 1B–D) that were found in the input sequence(s). Finally, one file contains a copy of the standard output. This latter file can be excluded using the --mute option. Temporary files can be kept using the --keep-tmp option. They include all the outputs from the different intermediate steps (prodigal, infernal, hmmer). These files can be very useful for certain tasks, notably if one wants to fine tune some parameters such as the aggregation distance threshold. In that case, IntegronFinder will not try to identify the *attC* sites or the integrase again, as they were already detected, but rather allows to only change the way the genetic elements are clustered. Since the temporary files take a significant amount of disk space, they are not kept by default. 

Some minor changes were made to the information outputted by IntegronFinder. CALIN are now reported when they have at least 2 *attC* sites (instead of just 1 as before). This value can be changed by the user with --calin-threshold. This is to diminish the probability of false positives when the CALIN are reported. Of note, CALIN with a single *attC* can be true positives, but we let the user decide what to do in that case. 

We also provide an alignment of *attC* sites to facilitate their analysis. Standard multiple alignment tools fail to align correctly palindromic sequences. Hence, IntegronFinder now outputs the alignment of *attC* sites built with Infernal, in which the different features of the *attC* sites (R and L boxes and unpaired central spacer) are correctly aligned. This alignment is available with the --keep-tmp option.

### 2.5. Availability

The code of IntegronFinder is available under the free and open-source license GPLv3 at https://github.com/gem-pasteur/Integron_Finder (accessed on 5 February 2022). The code has been packaged and is available on pypi https://pypi.org/project/integron-finder/ (accessed on 5 February 2022) to make it easy to install with the python packager installer *pip.* We also provide a bioconda package and a container solution https://hub.docker.com/r/gempasteur/integron_finder (accessed on 5 February 2022). A user-friendly Galaxy implementation can be found at https://galaxy.pasteur.fr/tool_runner?tool_id=toolshed.pasteur.fr%2Frepos%2Fkhillion%2Fintegron_finder%2Fintegron_finder%2F2.0%2Bgalaxy0 (accessed on 5 February 2022) [34]. The documentation is available at https://integronfinder.readthedocs.io/en/latest/ (accessed on 5 February 2022).

### 2.6. RefSeq Complete Genomes

The first dataset used in this study consists of 21,105 complete genomes retrieved from NCBI RefSeq database of high quality complete non-redundant prokaryotic genomes (ftp://ftp.ncbi.nlm.nih.gov/genomes/refseq/, last accessed on 30 March 2021) [35]. The complete list of genomes can be found in Appendix A. 

### 2.7. Klebsiella Data

The Klebsiella genomes were downloaded and annotated with PanACoTA [31] as explained in [36]. Briefly, we downloaded all the 5805 genome assemblies labeled as *Klebsiella pneumoniae sensu stricto* (*Kpn*) from NCBI RefSeq (accessed on 10 October 2018). We removed lower quality assemblies (L90 >100 and number of contigs >990) and strains that were too divergent from the reference strain (Mash distance > 0.06, [37]) or too similar (Mash distance < 0.0001) to other strains. The resulting set of 3980 sequences were consistently re-annotated with Prokka (v1.13.3) [38], and species assignments corrected with Kleborate (https://github.com/katholt/Kleborate, accessed on 5 February 2022) [39]. The accession numbers for the genomes are in Appendix A, along with all the annotations identified in this study. 

We built Klebsiella integron pangenomes using PanACoTA v1.3.1 pangenome module with default parameters (-i 0.8, -c 1). The latter clustered integron proteins (defined as the proteins encoded by genes lying between two *attC* or less than 200 bp away from the last *attC*) using MMSeqs2 13-45111 [40] with single-linkage and a threshold of 80% of identity and bidirectional coverage. 

We computed the species phylogenetic tree with IQ-TREE, as explained in [36]. Briefly, the 1431 protein families present in more than 99% of the strains in a single copy were aligned with Mafft (v7.407) [41], back-translated to DNA, and concatenated. We used this alignment containing 220,912 parsimony-informative sites over a total of 1,455,396 bp to infer a phylogenetic tree with IQ-TREE (v1.6.7.2) using ModelFinder (-m TEST) [42,43]. We used the best-fit model (GTR+F+I) without gamma correction, because of branch length scaling issues with our large dataset, and assessed the robustness of the phylogenetic inference by calculating 1000 ultra-fast bootstraps (-bb 1000) [44]. The mean support value was 97.6%. We placed the *Kpn* species root according to the outgroup formed by 22 misannotated *Klebsiella quasipneumoniae* subspecies *similipneumoniae* identified by Kleborate. 

### 2.8. Analysis of Antibiotic Resistance Genes (ARGs)

To annotate ARGs we used four databases: ARG-ANNOT (v5) [45], CARD [46], AMRFinderPlus [28] (v3.0.5) (v2019-09-07), and ResFinder (v2019-07-17) [47]. The annotations were made by searching for homology with Abricate (v0.9.8) (https://github.com/tseemann/abricate, accessed on 5 February 2022). While these databases are largely concordant, they do not all include the same types of functions. Hence, we compared them by looking at the number of databases reporting at least one hit (coverage > 80%, identity > 90%) for all the proteins identified in *Klebsiella pneumoniae* integrons. A large majority of proteins reported either 0 or 4 matches, but many had hits in only 1 to 3 databases. Noticeably, the second most abundant family in the pangenome, later labeled as efflux-pump multidrug transporters, hit only the AMRFinderPlus database. Hence, we decided to classify as ARG all the proteins reporting at least one hit in any of the four databases.

We classified each of the integron pangenome families as ARG or non-ARG, which resulted in homogeneous results within families (i.e., same number of databases reporting a hit), except for two families where a small fraction of genes (resp. 6% and 2%) had fewer hits than the other members. As a final criterion, we classified as ARG all the pangenome families for which at least 90% of the members had at least one hit in any of the four databases.

### 2.9. Saturation Curves

Pangenome saturation curves were computed with the function *specaccum* of the R package *vegan* with default parameters [48], calculating the expected (mean) gene richness using a sample-based rarefaction method that has been independently developed numerous times [49] and is often known as Mao Tau estimate.

### 2.10. Graphics and Visualizations

Graphics and visualizations were computed in Python with the library *seaborn* [50].

## 3. Results and Discussion

### 3.1. Distribution of Integrons across Bacteria

We used IntegronFinder v2 to search for integrons in the complete genomes of the RefSeq NCBI database, including 22,301 replicons named chromosomes and 21,520 replicons named plasmids (the few remaining were phages or not classed). The time that takes to run IntegronFinder is extremely dependent on the existence of *attC* sites in the genome. Genomes lacking *attC* sites run very fast, e.g., 3 s for a genome of *Bacillus subtilis* 168 with a genome size of ~4 Mb on a laptop. If the genome contains *attC* sites, then the use of the most accurate covariance model can slow down analyses, up to almost 4 min for the genome of *Vibrio cholerae* El Tor N16961 where there is one super-integron [12]. The entire database used in this study is almost ten times larger than the one used for the publication of version 1 (2484 genomes, [10]) and took around 150 CPU hours on a standard laptop. Given the difference in size to the previous database, we started by assessing integron distribution in the new much broader dataset (Figure 2A). We found 3116 genes coding for IntI and 56,994 *attC* sites. The classification of the elements by IntegronFinder v2 indicated the presence of 2760 complete integrons, 356 In0, and 2961 CALIN. These numbers are around ten times larger than those obtained 6 years ago, which is in rough proportion to the difference of size of the two datasets. However, we found a ratio CALIN/integron of almost 1, whereas it was almost 2 six years ago. This is caused by the change in the default parameters to identify CALIN that now require the presence of two *attC* sites. Among genomes that encode an integron, the frequency of In0 elements relative to integrons is also smaller in the new version. This is because the IntI protein is searched using appropriate thresholds (gathering thresholds) that diminish the rates of false positives, and this affects much more the counts of In0 than those of complete integrons (which also have neighboring *attC* sites and are therefore much less likely to be false positives). For example, 6 years ago, we identified two In0 in Actinobacteria. Since we found no complete integron in the clade, this suggested that our hits were false positives. We now find only one In0 (and still no complete integrons) with a much larger dataset that should have resulted in proportionally more false positives. 

Among genomes encoding at least one IntI, 22% encode more than one. This result confirms the observations made at the time of IntegronFinder v1.0, 6 years ago, where 20% of the genomes had more than one of the three [10]. One key difference between the current data and the previous data concerns the frequency of complete integrons encoded in plasmids. Six years ago, they accounted for a small minority of all the integrons detected, while they are now more numerous than those in chromosomes (53% of the total). The recent increase in the intensity of sampling and sequencing of nosocomial pathogens, which often have antibiotic resistance genes in mobile integrons (in plasmids), may explain these differences. 

We found an average of 14 *attC* sites per complete integron and almost 6 per CALIN. CALINs were proposed to be degraded integrons [10], and it is expected that they have fewer cassettes than integrons that encode an integrase allowing the integration of novel cassettes. We then separated integrons and CALIN with more than 10 *attC* (typically corresponding to sedentary elements) from the others. This shows, as expected, the presence of many more *attC* sites in complete sedentary integrons than in large CALIN (Figure 2B). When the analysis was restricted to clades known to have sedentary integrons (*Vibrio* and *Xanthomonas* spp.), we also found that CALIN tend to have fewer *attC* sites (Appendix A). Finally, as expected, integrons or CALIN in plasmids have fewer (respectively, 11 and 2.5 times less) *attC* sites than those in chromosomes. 

The distribution of integrons in plasmids revealed three elements with more than 10 *attC* sites (Figure 2C), which we studied in detail. The two largest elements were in very large replicons (1.1 Mb in *Gemmatirosa kalamazoonesis* KBS708 and 599 kb in *Vibrio* sp. HDW18) that are probably secondary chromosomes. Hence, the largest integron we could identify in replicons compatible with the typical sizes of mobile plasmids (1–300 kb, [51]) was a 10 *attC* integron in *Klebsiella* sp. RHBSTW-00464 plasmid 5 (46 kb). The second largest plasmid integron was in *Enterobacter hormaechei* subsp. steigerwaltii BD-50-Eh plasmid pBD-50-Eh_VIM-1 and was reported before [52]. Hence, while we had previously used 19 *attC* sites as a threshold to delimit mobile from sedentary integrons, our present results suggest that using a 10 *attC* sites threshold could be more adequate. 

The taxonomic distribution of integrons follows previously described trends [10,16,33]. The vast majority of complete integrons (93%) are found in γ-Proteobacteria, a percentage that largely exceeds the representation of these genomes in the database (38%) (Figure 2D). In line with previous work, we did not identify integrons in *Chlamydiae* nor in *Tenericutes*, despite the good representation of these two phyla in our dataset (respectively, 170 and 400 genomes). However, the much larger dataset analyzed in this work, as well as some changes in taxonomy and species classification, led to the identification of several clades that lacked integrons six years ago and now have a few. For example, we found 2 CALIN elements in the phylum Nitrospira, both occurring in the genome of *Candidatus Nitrospira inopinata*, a bacterium known to perform complete ammonia oxidation to nitrate [53]. The newly defined phylum Acidithiobacillia has 2 complete integrons and 8 CALIN over a total of 11 genomes. Finally, while only In0 had been detected in Bacteroidetes in our previous study, we identified complete integrons in the Bacteroidetes/Chlorobi group. More precisely, on top of the ones found previously in Chlorobi and Ignavibacteriae, we identified a complete integron in *Salinibacter ruber* M1 [54], a halophilic bacteria that belongs to Bacteroidetes. 

Our previous study failed to identify complete integrons in Actinobacteria, Firmicutes, and Alpha Proteobacteria. We now identify a few elements in these clades. However, their frequency is extremely small relative to the high frequency of such genomes in the database. Firmicutes account for 22% of the genomes, Alpha Proteobacteria for 6%, and Actinobacteria for 10%. The only complete integron in Firmicutes was found in *Limnochorda pilosa* strain HC45, and a few CALIN were found in different genera across the phylum. The 6 complete integrons in Alpha Proteobacteria were found in species of Sphingobium and Agrobacterium. As mentioned above, Actinobacteria lack complete integrons, have only one In0, and a few CALIN. These results show that while integrons, or their components, are not altogether missing in these clades, in line with previous works [55,56], they are indeed extremely rare and the observed occurrences may represent recent acquisitions. Considering the presence of integrons in very diverse clades and their ability to spread within plasmids and transposons, the lack of integrons in certain clades remains intriguing as it suggests the existence of some incompatibility between integrons and the genetic background of these large bacterial clades. 

A recent preprint revealed the presence of integrons in metagenome assembled genomes (MAGs) of Archaea, but not in complete genomes [25]. Since this study used a combination of a pre-release of IntegronFinder 2.0 and HattCI [24], we wished to understand if the novel version of IntegronFinder alone, which is better suited to study MAGs, could identify these systems. We also failed to identify complete integrons, CALIN, or In0 in the genomes of Archaea present in our database. This is not unexpected since the clades where the integrons have been identified are very poorly represented in RefSeq. Therefore, we downloaded five MAGs of the study cited above and searched them for integrons. We confirmed the presence of two complete integrons, one CALIN, and one In0 whereas one complete integron in [25] was detected as a CALIN in our analysis. Hence, if Archaea are found to encode integrons like those suggested by the recent analysis of MAGs, IntegronFinder version 2 is expected to be able to identify them. 

### 3.2. Recent Spread of Integrons in K. pneumoniae

To exemplify the use of the novel version of IntegronFinder, we analyzed a large collection of draft genomes of *Klebsiella pneumoniae* (*Kpn*). *Kpn* is a Gram-negative bacterial species belonging to the Enterobacteriaceae family, naturally occurring in soil, freshwater, and mammalian gut and is considered an opportunistic pathogen [57]. It is a member of the ESKAPE pathogens, the list of high-risk multi-resistant nosocomial pathogens from the World Health Organization. Aside from having the largest known mobile integron in the RefSeq database (see above), *Kpn* is also an interesting example of a nosocomial bacteria that is thought to have recently acquired integrons carrying antibiotic resistance genes. While the presence of integrons in the species is well known [58], its characterization has not been done at the species level. Hence, we collected 3980 genomes of the species, re-annotated them uniformly, and searched for complete integrons in the draft genomes. The analysis by IntegronFinder took 14 h using 5 CPUs on a standard desktop computer.

We identified 1855 complete integrons, 2590 CALIN, and 405 In0 in 2709 of the 3980 genomes, resulting in a (CALIN+In0)/complete ratio of 1.61 (Appendix A). As our dataset mostly comprises draft genomes, it was expected that some complete integrons would be divided in several contigs, leading to an increase of the proportion of CALIN and In0. This phenomenon may be enhanced by the presence of *attC* sites very similar in sequence, which complicates the assembly of contigs harboring integrons. For comparison, among the 730 *K. pneumoniae* complete genomes comprised in our database extracted from RefSeq (see previous section), 492 complete integrons, 282 CALIN, and 21 In0 were identified in 435 genomes. The (CALIN+In0)/complete ratio is 0.62 in this dataset, which seems to confirm that analyses of draft genomes will tend to overestimate the frequency of In0 and CALIN. To further investigate how many complete integrons may have been split between contigs, we looked for occurrences of In0, complete integrons, and CALIN where the integrase or a gene cassette was the first gene in a contig. We found 819 isolates harboring at least one CALIN together with one complete integron or In0 at the border of a contig. Among these, we found 339 In0 and 532 complete integrons. These might correspond to true complete integrons that were split between contigs during assembling. Hence, the analysis of genomes that are not fully assembled may result in the underestimation of the number of cassettes, incomplete integrons, and spurious identification of CALIN elements.

For the following analysis, we decided to focus on the 1855 complete integrons identified in 1677 genomes. In other words, around 42% of the genomes of our *Kpn* in the database have an integron (all data in Appendix A). In genomes carrying integrons, the number of integrons per genome is usually one but can be higher. Notably, the genomes of strains AR_0039 (GCF_001874875.1) and DHQP1002001 (GCF_001704235.1) carry four complete integrons each. To study the distribution of integrons across the species, we built a phylogenetic tree and plotted the presence of integrons in function of the sequence type (ST) (Figure 3). This analysis revealed no clear pattern of aggregation, since integrons are found across the species. This suggests multiple independent acquisitions of integrons in the species recent past.

Integrons in *Kpn* have been studied because they have a few well-known antibiotic resistance genes [59,60]. One might thus have expected to find little genetic diversity among them. To assess the diversity of these integrons, we analyzed their gene repertoires. The integrons in *Kpn* encoded a total of 5763 protein coding genes, with the largest elements having up to 15 genes or 7 *attC* sites. Hence, we could not find in this dataset integrons as large as the one found in the complete genomes of RefSeq. This may be caused by different samplings or because integrons in draft genomes may be split in several contigs. We used PanACoTA [31] to compute the pangenome of the integrons and identified 165 different gene families. Hence, the integrons of *Kpn* are small, as expected from mobile integrons, but carry a large diversity of gene families. We computed saturation curves for the integrons’ pangenome and observed that it is open, i.e., after analyzing almost 2000 elements, the curve does not show evidence of saturation (Figure 4). Further genomic sampling of *Kpn* will thus likely reveal novel gene cassettes.

We then detailed the functions of the genes encoded in the cassettes of integrons (IntI being obviously the most frequent protein of integrons). Expectedly, most of the largest gene families in cassettes are associated with antibiotic resistance genes (ARG). To study them in detail we used four databases of ARGs: CARD, ARG-Annot, ResFinder, and AMRFinderPlus. The integrated analysis of the results (see Methods) resulted in the identification of 50 families of ARG. An analysis of the 20 most frequent gene families reveals 12 that are ARGs, including multidrug transporters and enzymes associated with resistance to antiseptics, aminoglycosides, diaminopyrimidine (e.g., trimethoprim), fluoroquinolones, rifamycin, chloramphenicol, or beta-lactams. The second largest family of the cassettes codes for the protein AadA2, an aminoglycoside nucleotidyltransferase, and is present in more than a third of the integrons (654). This family is also present in multiple copies within an integron in 50 genomes. The list of resistances to beta-lactams was very diverse, including OXA-1, OXA-2, OXA-10, OXA-18, and metallo-beta-lactamase. To assess the diversity of ARG families, we computed a saturation curve for this fraction of the pangenome (Figure 4). The results show that while this subset is necessarily less diverse, it still accounts for up to 50 different gene families, and that further sampling of *Kpn* genomes will most likely reveal novel genes in extant *Kpn* populations. Some of the other large families encode functions implicated in recombination, notably DDE or other types of recombinases. However, most families of genes are of unknown function, suggesting that even in species such as *Klebsiella*, where integrons seem to be recent and driven by selection for antibiotic resistance, there may be many other relevant functions encoded in integrons.

## 4. Conclusions

Many of the improvements in IntegronFinder version 2 will be almost invisible to the user because they involve software engineering, but they will facilitate its maintenance in the next decade. This is important because funding for software maintenance is almost inexistent. The novel most visible capacities of IntegronFinder will allow it to better tackle large datasets of genome, which tend to be composed of draft genomes, and metagenomes. Here, we have exemplified its use with the database of complete genomes and a large dataset of *Kpn* draft genomes. As it stands, we regard speed as a major limitation to the use of the program in exceptionally large datasets, e.g., in metagenomes or in datasets with hundreds of thousands of genomes. The current deadlock is the use of the covariance model that is computationally expensive in its most accurate version. Significant advances in the speed of IntegronFinder will require the development of novel methods to use such models.

## Figures and Tables

**Figure 1 microorganisms-10-00700-f001:**
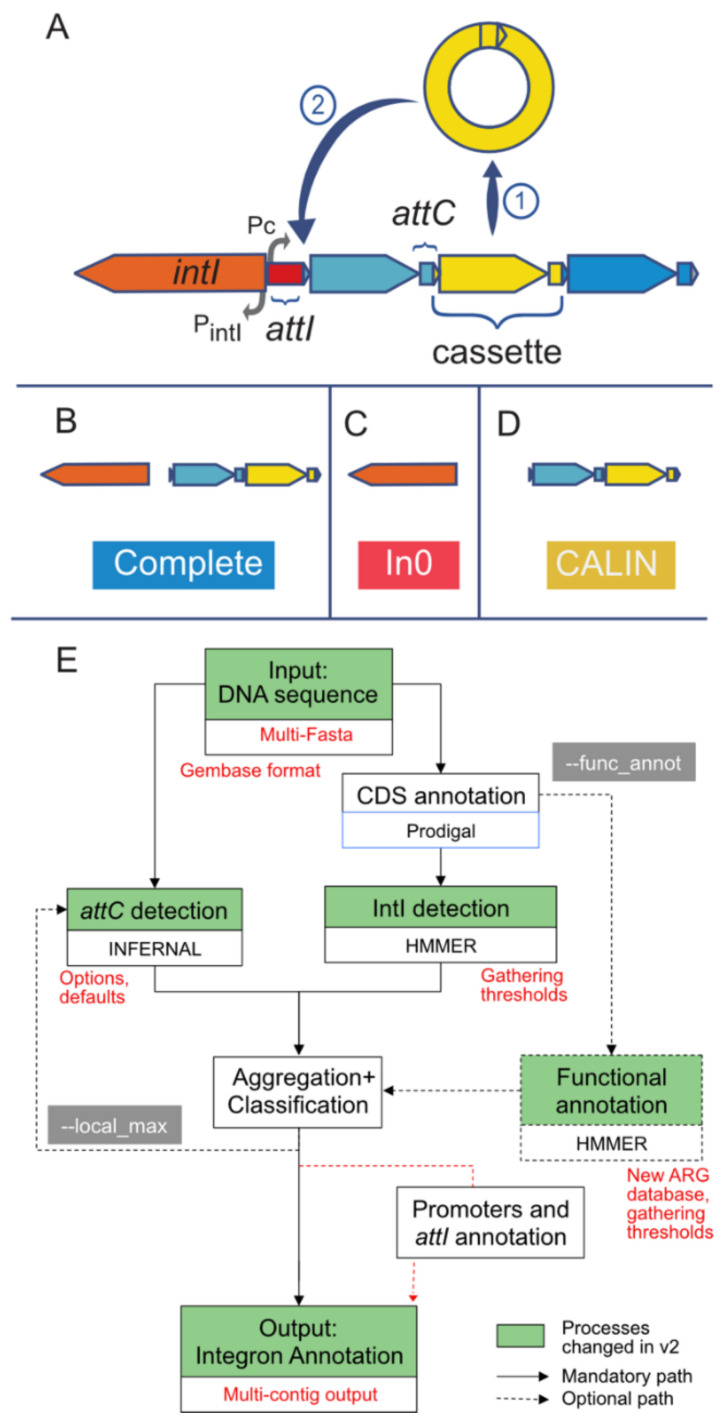
(**A**) Key genetic elements of integrons. (**B**–**D**) Different types of elements searched by IntegronFinder v2. (**E**) Diagram describing the different steps to identify and annotate integrons with in green the processes that were changed in some way (changes in red). Figure modified from [10].

**Figure 2 microorganisms-10-00700-f002:**
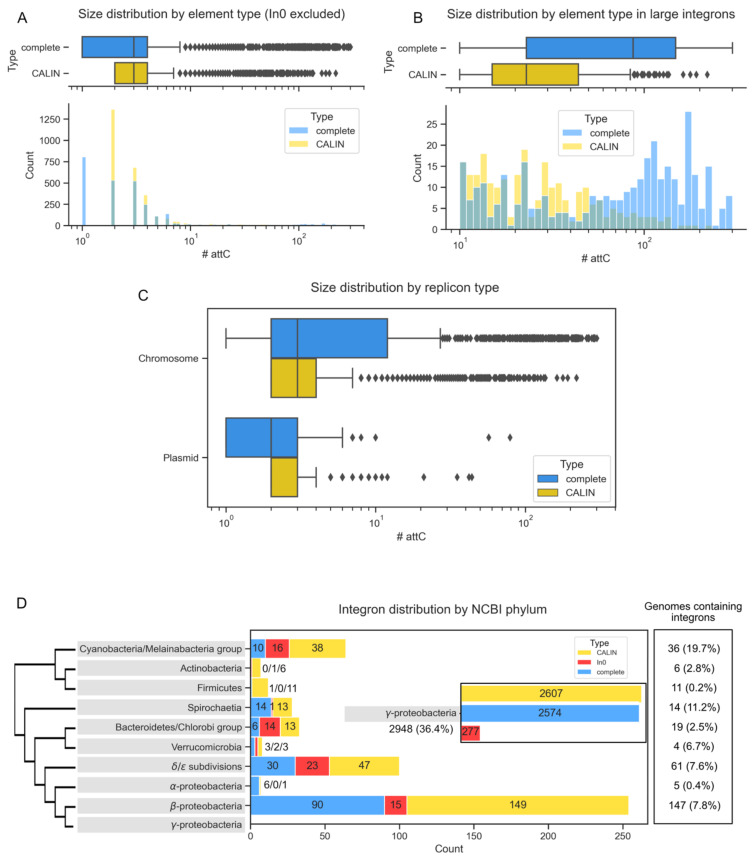
Statistics concerning the distribution of integrons in the RefSeq NCBI database. Distribution of the number of *attC* sites found per element (complete integron or CALIN) (**A**) and zoom on the distribution of large elements (>10 sites, **B**). Distribution of number of *attC* sites per type of integron and replicon (**C**). Distribution of integrons across major bacterial phyla. Only phyla comprising integrons are shown. The percentage in the last box is the proportion of the genomes in our database that contain at least one integron (**D**).

**Figure 3 microorganisms-10-00700-f003:**
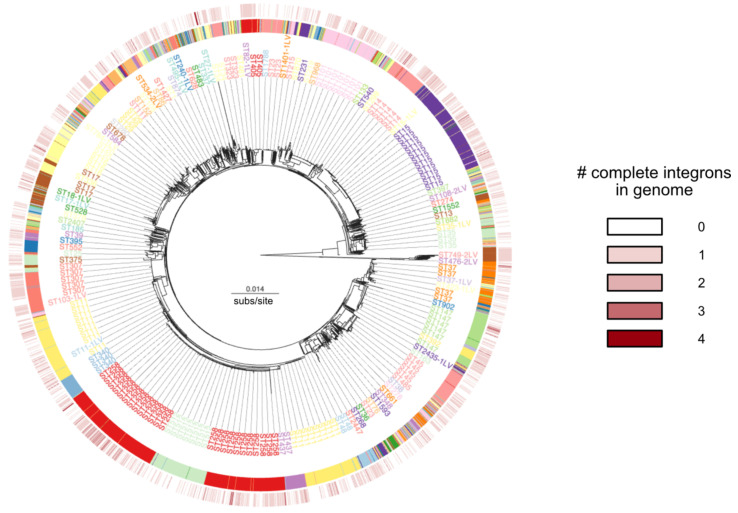
Phylogenetic tree of the core genome of *Klebsiella pneumoniae*, with an indication of the sequence type (ST) in the ribbon (and mention to the most frequent in the intermediate circle). The outer circle indicates the number of complete integrons in each genome. Tree was built as indicated in Methods and drawn using Microreact [59].

**Figure 4 microorganisms-10-00700-f004:**
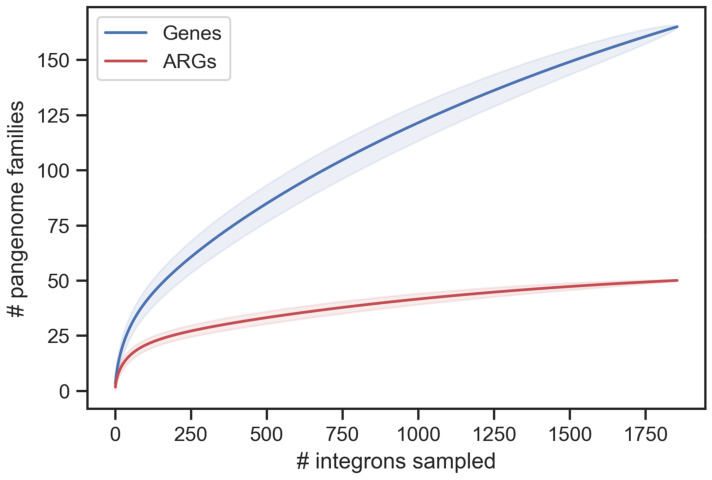
Analysis of the pangenome of the integrons of *Kpn*. The figure displays two saturation curves, representing the expected (mean) gene richness when increasing the number of integrons. They were computed using the Mao Tau estimate (see Methods for more details). The shadowed regions correspond to one standard deviation.

## Data Availability

The raw data of this study will be made available by the authors, without reservation, to any qualified researcher. The software code is publicly available in the Git.

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
