# Peer review of "IntegronFinder 2.0: Identification and Analysis of Integrons across Bacteria, with a Focus on Antibiotic Resistance in Klebsiella"

_microorganisms, 2022, doi:10.3390/microorganisms10040700_

Round 1
Reviewer 1 Report
Manuscript ID microorganisms -1643943 Neron et al.
IntegronFinder 2.0: identification and analysis of integrons across Bacteria, with a focus on antibiotic resistance in Klebsiella
Integron Finder is a very useful and widely applied tool to extract integron sequences from genomic data. Here the authors have implemented a number of improvements to the platform, in coding, in inputs and in outputs. They have improved the functionality and applicability of the program. Changing the reporting of CALINs to include only those with two attC sites is an improvement that increases the stringency of reporting.
The improved version has been tested using genomic data, focussing on the incorporation of draft genomes, MAGs and metagenomes. The application of the new version has been ground tested using the available genomes of Klebsiella. This analysis shows some important and interesting results. The differences between saturation in cassette encode ARGs and other genes in integrons is intriguing and important. It suggests that some of the non-ARG cassettes confer phenotypes of relevance to transmission, survival and pathogenicity.
Integron Finder will continue to be a valuable tool for investigations of genome structure, bacterial evolution and horizontal gene transfer.
I have no substantive comments on the content or conclusions made in the MS.
Some minor comments in order of appearance in the MS
Page 1
• ‘data’ is plural, so for instance in the Abstract, it should be ‘Our data show….’
• The first part of the Introduction is not a sentence – change to ‘Bacteria evolve novel traits by many processes, including horizontal gene transfer (HGT) driven by phages, conjugative elements, or natural transformation.'
• that allows to IntI to be easily distinguished from other integrases [8].
Page 2
• The cassettes are flanked by another type of site (attC)
• I don’t think that ‘(now deprecated)’, which means to ‘express disapproval of’ is the sense of what you mean. Perhaps ‘superseded’? Replace further down (page 3) as well.
Page 4
• You say that attI sites are rarely found outside class 1 integrons, but what you mean is that they are poorly defined and hard to detect in other integrons. The attI site in class 1 integrons is called attI1. Other integrons must have an analogous, if not homologous site for insertion of incoming gene cassettes. The wording here is ambiguous, so phrase for clarity.
Page 5
• “IntegronFinder may not search to identify the attC sites or the integrase again’
Page 6
• The ‘persistent genome’ is usually called the ‘core genome’
Page 8
• ‘found in different genus’ change to genera
• The final paragraph outlines results from a preprint on the discovery of integrons in Archaea. This study did use Integron Finder 2.0, not the first version.
Page 9
• ‘multi-resistant nosocomial pathogens’
Page 10
• ‘2709of the 3980’ insert space after 2709
I congratulate the authors for their ongoing maintenance of this valuable tool, which we have found to be very useful in our own research,
Prof Michael Gillings
Reviewer 2 Report
In this manuscript the authors present an upgrade in the IntegronFinder a software available at https://github.com/gem-pasteur/Integron_Finder and analyze Klebsiella pneumoniae genomes for the presence of integrons and antibiotic resistance genes using this platform. In the study, particularly concerning the biological/genetics concepts, is possible to see confuse/unsounded statements as well as inadequate references. Would be better to present it as a technical note.
In this way, below it is shown some examples of such situations that have to be rewritten/modified:
In the Abstract and Introduction for example is stated: “Integrons are mobile genetic elements” (abstract), “One usually distinguishes sedentary from mobile integrons [6] (introduction)” but integrons are not mobile elements. Eventually, when they part of mobile elements such as plasmids, transposons…. . there is a “mobilization”.
Introduction: “Class I integrons are particularly abundant and have the capacity of recombining more diverse types of cassettes than sedentary integrons [14]” has to be rewritten: what do you mean with “diverse types”? sedentary integrons presents a huge diversity of cassettes
The authors have to check the references in all text, for example:
This is a limitation since recent studies have uncovered novel integrons in metagenomes and in metagenome assembled genomes [31, 32].
- Haudiquet M, Buffet A, Rendueles O, Rocha EPC. Interplay between the cell envelope and mobile genetic elements shapes gene flow in populations of the nosocomial pathogen Klebsiella pneumoniae. PLoS Biol. 2021;19:e3001276.
- Katoh K, Standley DM. MAFFT multiple sequence alignment software version 7: improvements in performance and usability. Mol Biol Evol. 2013;30:772-80.
Are you sure that these references support/is linked to your sentence?
Results/Discussion
Some unsounded sentences have to be rewritten and reference(s) added:
“Among genomes encoding at least one integron/CALIN/In0, 22% encode more than one. This result confirms the observations made 6 years ago (20%). One key difference between the current data and the previous one concerns the frequency of complete in-tegrons encoded in plasmids. Six years ago they were a small minority whereas now they are more numerous than those in chromosomes (53% of the total).”
Recent spread of integrons in K. pneumoniae
The results in this entire sub-section are quite obvious (in some cases) ant too speculative in others. Therefore, keep your focus on the main finding.
